

# The millennium old hydrogeology textbook "The Extraction of Hidden Waters" by the Persian mathematician and engineer Abubakr Mohammad Karaji (c. 953 – c. 1029)

Behzad Ataie-Ashtiani[1,2], Craig T. Simmons[1]

[1] National Centre for Groundwater Research and Training and College of Science & Engineering, Flinders University, Adelaide, South Australia, Australia
[2] Department of Civil Engineering, Sharif University of Technology, Tehran, Iran

*Correspondence to*: Behzad Ataie-Ashtiani (behzad.ataieashtiani@flinders.edu.au)

**Abstract.** We revisit and shed light on the millennium old hydrogeology textbook "The Extraction of Hidden Waters" by the Persian mathematician and engineer Karaji. Despite the incomparable understanding and conceptualization of the world by the people of that time, ground-breaking ideas and descriptions of hydrological and hydrogeological perceptions such as

components of hydrological cycle, groundwater quality and even driving factors of groundwater flow were presented in the book. Although some of the ideas may have been presented elsewhere, this is the first time that a whole book was focused on different aspects of hydrology and hydrogeology.  More importantly, we are impressed as the book is composed in a way that covered all aspects that are related to an engineering project including technical and construction issues, guidelines for maintenance, and final delivery of the project when the development and construction is over. We speculate that Karaji's book

is the first of its kind to provide a construction and maintenance manual for a modern engineering project.

## 1 Prologue

The eleventh century Arabic book "Inbat al-miyah al-khafiya" (Persian: انباط المیاه الخفیه) or The Extraction of Hidden Waters by Abubakr Mohammad Ebn Al-Hassan Al-Haseb Al-Karaji (Persian: ابوبکر محمد بن الحسن الکرجی) is a pioneer text on hydrogeology (Karaji, 1941). The book is in Arabic, the scholarly language of Persia in the Medieval Islamic Civilization era.

The book was translated from Arabic into Persian by Hoseyn Xadiv Jam in 1966 (Xadiv Jam, H. 1966).  Karaji's book was also translated into French in 1973 (Mazaheri, 1973), Italian in 2007 (Ferriello, 2007), and into English in 2011 by Schade in the PhD Thesis (Schade, 2011). Schade's translation was from the French translation to English.

Nadji and Voigt (1972) presented a glimpse into the book. In their interesting article, they stated that, based on this 11th century book, the basics of hydrologic cycle and components of underground water quality were already known by the Arab and

Persian scientists of that time. They mentioned that the techniques of wells and qanats digging that were developed for groundwater exploitation in the Middle East were of such a high standard that they are still in use today. Prompted by Nadji and Voigt (1972), Davis (1973) put Karaji's work in a broader scientific context, explaining the lack of appreciation, value and awareness of Middle Eastern science and scientists in general terms.



We believe that Karaji's contributions in hydrology and hydrogeology are significant and should be remembered and revisited

in this Hydrology and Earth System Sciences special issue on the 'History of Hydrology'. In this essay, we revisit this book and provide an English translation of the pieces from the book that crucially offer pioneering ideas in hydrogeology and in general for engineering projects. The translations presented here are based on the Persian translation of Karaji's book originally written in Arabic. We believe it is crucially important to include quotes from Karaji, in his own words, to ensure historical veracity and authenticity and hence a historically faithful essay. It is also fascinating to hear Karaji's thoughts in his own words

– bringing his story, his motivations and his scientific contributions to life.

We hope this essay brings about new insights and information that were not provided in the previous written accounts. We hope that it helps to contribute to a growing awareness of Karaji's contributions to hydrology. In the following sections, we provide a short description of  Mohammad Karaji's life, explanations of basic components of Qanat technology to exploit groundwater resources, and finally examine Karaji's book "The Extraction of Hidden Waters" to shed some light on his

knowledge of hydrology and hydrogeology some one thousand years ago.

### 1.1 Abubakr Mohammad Karaji

Abubakr Muhammed Karaji was a late 10th century-early 11th century (c. 953 – c. 1029) Persian-born Muslim mathematician and engineer. Most of his scientific life was spent in Baghdad.  Girogio Levi Della Vida (1934) mentioned he was born in Karaj, a city near Tehran, Iran, and was not from Al-Karkh district of Baghdad, Iraq (Abattouy, 2019). Karaji lived in Baghdad

under the Abbasid rulers. We anticipate that he would have been a direct beneficiary of the translation movement. This initiative was begun under the second Caliph Al-Mansur and continuing through to the seventh Caliph Al-Ma'mun and saw a large amount of significant scientific, religious and other literature translated into Arabic for scholars to use. At this time, Baghdad was one of the world's greatest places of learning and knowledge. It hosted some of the world's best libraries. It was a vibrant place for scholarly activity and scientific discovery. The Middle East became the centre of intellectual thought instead

of Europe. The modern world owes a great deal to the far thinking translation initiative of the Abbasid Caliphate and generally to Muslim (Arabic-Persian) Golden Era civilization.

Karaji lived in Baghdad for most of his life and his main mathematical works were written when he lived in that city (O'Connor and Robertson, 2019). His three remaining books on algebra and geometry are: Al-Badi' fi'l-hisab (exquisiteness of calculation), Al-Fakhri fi'l-jabr wa'l-muqabala (Glories of  algebra), and Al-Kafi fi'l-hisab (Sufficient for calculus) (Abattouy,

2019). The titles of his books on mathematics signals Karaji's lookout on mathematics. It portrays his affection for mathematics as a spectacular and almighty knowledge. In the introduction of Xadiv Jam's translation where, a historical account of the life and work of Karaji was presented,  it was mentioned that Karaji was a contemporary of great Persian scholars such as Avicenna (c. 980 – June 1037), Al-Biruni (c. 973– c.1050) and Al-Razi (c. 854– c. 925).

A short historical perspective of Karaji's importance in the development of mathematics is given at MacTutor History of

Mathematics archive (O'Connor and Robertson, 2019). Based on O'Connor and Robertson (2019), Woepcke (1853) described the importance of Al-Karaji's work on the first appearance of a "... *theory of algebraic calculus* ...". Also, Rashid (1994) wrote





*"Al-Karaji's work holds an especially important place in the history of mathematics. ... the discovery and reading of the arithmetical work of Diophantus, in the light of the algebraic conceptions and methods of al-Khwarizmi and other Arab algebraists, made possible a new departure in algebra by Al-Karaji ...".*

Karaji described a binomial coefficients theorem similar to the Pascal triangle (O'Connor and Robertson, 2019). Abrarova (1984) described some of Karaji's contributions in geometry. Karaji defined points, lines, surfaces, solids and angles, gave rules for measuring both plane and solid figures, and provided methods of weighing different substances (O'Connor and Robertson, 2019).

In later years of his life, Karaji returned to the central plateau of his Persian homeland and wrote the book Inbat al-miyah al-
khafiya ("The Extraction of Hidden Waters)". This book was about practical hydrology in this period. Although, it has been mentioned that the book was written by him as a means of earning a living, we may speculate that the topic was of great practical interest in the arid area of the Persia plateau. It is also very likely that this topic was of interest to Karaji personally and he certainly knew it was vitally important. As it will be mentioned, in the extracts of Karaji's preface to his book, he notes that to provide people with a good water supply would be a most beneficial work. The book is considered "the oldest textbook
on hydrology" (Nadji and Voigt, 1972). Figure 1 shows a statue of Abubakr Mohammad Karaji at the Water Museum of Sa'dabad Museum Complex in Tehran, Iran.

## 1.2 Qanat

Karaji wrote extensively on Qanat in his book. Qanat or Kariz is an old system of water supply from an aquifer. Qanat is an Arabic word and Kariz is in Persian, although Qanat is now also used in Persian. It consists of gently sloping underground
aqueducts, which are hand-dug and just large enough to fit the person doing the digging, and a series of wells and vertical access shafts traversing different topographies and geologies along its course [e.g., English, 1968; Semsar Yazdi and Labbaf Khaneiki, 2017]. Vertical shafts are used to remove excavated material, to ventilate tunnels, and to provide access for maintenances. Qanats are still used in arid and semi-arid climates for the supply of water. Qanat technology was developed for the first time in ancient Persia as far back as 3000 BC to the early 1st millennium BC [e.g., Korka, 2014; Hussain et al.,
2008; Wulff, 1968].

Qanat technology spread across the world, first westwards to the Mediterranean and Egypt, and southwards to Oman and Southern Arabia, and then the second major diffusion of Qanat technology occurred with the early conquests of Islam to peninsular Spain and the Canary Islands [Lambdon, 1989; Martínez-Santos and Martínez-Alfaro, 2012]. Finally, as a consequence of the Spanish conquests, the technology also spread to South and Central Americas, such as in Mexico, Peru, or
Chile [Martínez-Santos and Martínez-Alfaro, 2012].

Karaji's book not only explains his understanding of the hydrology at his time, but it also provides a practical manual on how to construct Qanat.



## 2 The Extraction of Hidden Waters

In the preface to the book Karaji wrote "*I do not know any other profession more beneficial than extraction of hidden water,*
*as it flourishes and cultivates lands, improves people's welfare, and grants ample profits*" [Translated from Xadiv Jam, H.
1966]. Figure 2 illustrates the first page of Inbāt al-miyāh al-khafiya. This is from a later-century copy of the original book of
Karaji that is kept at the University of Pennsylvania, in the Lawrence J. Schoenberg Collection [Karaji, 1675].

Section titles in the book, in Karaji's own words, are: the earth, about hidden waters; the mountains and rocks that indicate
water; the lands that indicate water presence; the plants that indicate water presence; about arid mountains and lands; types of
water and their tastes, distinguishing water qualities (heavy, light, thick, thin, potable and undrinkable waters); remediation
methods for contaminated water; about seasons, about land soils; about the protection zone of wells and qanat based on
religious rules; about water flow in qanat gallery (channel) segment (*Tanbooshe*); about the slaked lime cement for connecting
segments, preparations for water flow without *Tanbooshe* installation; about application and the invented surveyor's level
tool; measurement of mountains heights, the construction of Qanats; about reinforcement of underground tunnelling
excavations; about excavation methods in irregular tunnels; on the maintenance of Qanats; dealing with blockages; about
taking the project from excavators (Xadiv Jam, H. 1966).

The titles of the book sections provide a fascinating insight into the wide range of topics that were covered in the book. It is
amazing that the book not only covers the conceptual and technical aspects as well as construction guides, it also provides
guidelines for maintenance and even advice on how to deliver and consign the project when the development and construction
is over. It even touches on important social aspects such as religious regulations. The book is like a construction and
maintenance manual for a modern engineering project!

Excerpts from Karaji's book highlight his knowledge of hydrology at the time:

" … *Earth with all its mountains, plains, low, and high lands, is spherical form*…" [Translated from Page 24: Xadiv Jam,
1966]. Karaji believed that each component of the universe such as fire, air, water and soil have a specific location and intends
to get back to their original location when they separate from their source. "… *therefore, water flows from distant to closer*
*locations from earth centre, and by transformation/conversion of air to water in cold days and cold locations and conversion*
*of water to air in hot seasons and warm locations this flow continues and this transformation of water and air to each other is*
*very beneficial for earth affluence.* " [Translated from Page 26: Xadiv Jam, 1966]. Obviously, those who lived a millennium
ago, had a very different understanding and conceptualization of the world surrounding them. It should be considered that the
classical elements air, earth, fire and water were used by medieval scientists to explain nature.

"*I have heard that in some islands there are excessive freshwater springs, and there is no doubt that that the source of them is*
*not the surroundings seawater of islands, as the seawater level is lower that the island surface level and seawater is brackish*
*but the springs' water are fresh. However, the sources of these springs are distant locations that have higher level than the*
*springs' level*…" [ Translated from Page 29: Xadiv Jam, 1966].



"*And a portion of water that infiltrates into ground, when it reaches to hard soil, it avoids infiltration and rests there. And when aqueducts are established above these barriers, water enters into these conducts proportional to its force and pressure.*" [Translated from Page 32: Xadiv Jam, 1966].

Karaji referred to the importance of water quality and taste and the possible causes of water quality deterioration. "*I saw a river flowing in a valley near a village called Kandeh adjacent to Saveh and its water was fresh. There was rock with three*

*openings inside of the river and drinking the bitter water flushing out of the openings would cause diarrhea. Without any doubt the source of that water was not the rock and the river water, however, this water infiltrated into the ground somewhere far from the rock and flowing into the soils it has passed through in its path caused the change of the water's taste.*" [Translated from Page 32: Xadiv Jam, 1966].

Karaji provided some text on the sources of water and a preliminary indication of the hydrologic cycle. "*And God created*

*water in a way that it fills most of earth's cracks and fissures, and its surplus overflows into sea. Thus, the source of most water is snow and rain and transformation of water into air and air into water…*" [Translated from Page 34: Xadiv Jam, 1966]. Based on this quote, and the textbook more generally, we assert that Karaji essentially understood the crux of the hydrologic cycle as we know it today. To appreciate the significance of Mohammad Karaji's 1000 year old book and his working knowledge of hydrology, it is important to compare the Middle Ages European knowledge of hydrology. The basic

principal of hydrology and correct representation of the hydrologic cycle were represented by Palissy (1509-1590), a French scientist and potter, some five or six hundred years after Karaji (Duffy, 2017).

Then Karaji explains the procedure to extract fresh water from the sea floor. "*…seawater is heavy and undrinkable, as sunlight takes its thinness and freshness during long time. The evidence for this proposition is that sailors exploit and drink freshwater from the sea floor.*" [Translated from Page 38: Xadiv Jam, 1966]. The freshwater mentioned at the sea floor is possibly due to

the discharge of offshore fresh groundwater that is now well known as being a global phenomenon we today refer to as submarine groundwater discharge [Post et al., 2013].

Karaji provided observations and evidence which can be considered to describe groundwater-surface water interactions in today's nomenclature. "*…that water in the wells rises when water in rivers increases and falls when that decreases, to the extent that the water level in a well would be the same as the water level in a river*" [Translated from Page 40: Xadiv Jam,

1966]. "*…and the rainwater infiltrates into earth openings and gaps till water encounters a horizontal barrier and stops there.*" [Page 41: Xadiv Jam, 1966]. It seems as an understanding of recharge processes and the way in which water interacts with rocks – earliest conceptions of "hydrogeology" – water and rock.

Karaji provided explanations about soils and rock classifications based on their colours and characteristics and described the indicators that could be used to find out where water might be available underground and in springs. One of the indicators

Karaji stated could be usefully employed is lush land and the ampleness of vegetations and trees – indicators of the potential dependence of vegetation and ecosystem health on groundwater – what we call groundwater-dependent ecosystems today. He even specified the type of plants in this regard based on observations and reliable narratives. Simmons (2008) wrote about Father Paramelle as a naturalist who published "The Art of Discovering Springs" the same year as Darcy (1856) and the



publication of Darcy's Law. Paramelle's work was the best seller not Darcy's. Darcy disliked Paramelle's works to begin with

but eventually came around to see the usefulness in Paramelle's observations and recognised him as a good geologist concerned with underground hydrography (Simmons, 2008). Fascinatingly, Paramelle provided similar observations to Karaji, about 800 years later.

Influence and interaction of soil and vegetations on the water passing through them. "*And snow water and rainwater are the most delectable water, and afterwards the water that flows over impeccable soils or over sand and fine stone pieces, and in*

*channels without any vegetation. The taste of other water, that does not have these features, would change by the soil and vegetation in their path*." [Translated from Page 50: Xadiv Jam, 1966]. Karaji explained about water quality and important sanitary matters, and the possible illnesses caused by unhealthy water based on water taste, odour, weight and temperature. He also proposed some methods to treat brackish and unhealthy water. "*... whenever in a container of brackish or heavy water clean and neat ground soil would be added and then put the container aside till water is still and clear, some part of salinity*

*and heaviness would be removed. If this procedure is repeated water gets improved; and if this water is poured into a new pot till water leaks and drops from its bottom, a portion of salinity and heaviness is removed*" [Translated from Page 53: Xadiv Jam, 1966]. The treatment Karaji outlined is essentially a water filtration process based on the knowledge and apparatus of the time.

Karaji went on to provide explanations about different seasons and their influences on water quantity. He provides a brief

outline of climatology knowledge of the time. He wrote about different types of soils and their influences on the stability of the excavated Qanat. Karaji described methods and measures to find the location of water underground. For example, "*If there are dry pits or wells and we want to know if there any water there or not, a piece of dry or oiled wool which in connected to a string is dangled into the well, if the wool does not reach to the bottom of well and does not touch the well's wall, and it is suspended for three hours in this situation and it is taken out after that, if there is moisture in the wool then there is water in*

*that place.* " [Translated from Page 61: Xadiv Jam, 1966]. He explained about earthquake effects of groundwater flow. "*once an earthquake occurs springs gush and sometimes new springs appear, or the location of springs are displaced.*" [Translated from Page 61: Xadiv Jam, 1966].

Karaji described water flow in Qanat and wells. "*Of course, it is not possible that water of a spring or well or lagoon gushes or rises up, unless its source is in location that is higher than the location of gushing.*" [Translated from Page 63: Xadiv Jam,

1966]. Concepts such as mass, force, energy, gravity field, and may other physical properties and processes, which are easily comprehensible now, did not exist in eleventh century conceptualizations of the universe. However, we may speculate that there are some very early insights into the modern-day concept of hydraulic head – namely that groundwater flows from points of high hydraulic head to points of low hydraulic head – in Karaji's descriptions of water flow. We are unaware of any other documented cases where ideas of groundwater flow, from higher grounds to lower grounds, had been published any earlier

than Karaji's treatment.

To understand how different the conceptualisation of the world was in old times, the following is a quote from da Vinci (1452-1519) to explain water flow, in which he creates an analogy between water flow and blood circulation in the human body:



*"Natural heat keeps blood in the veins at the top of the man, and when the man has died this blood becomes cold and is brought back into the low parts, and as the sun warms the man's head the amount of blood there increases, and it grows to such an*

*excess there with the humors as to overload the veins and frequently to cause pains in the head.*

*It is the same with the springs that ramify through the body of the earth and, by the natural heat which is spread through all the body that contains them, the water stays in the springs and is raised to the high summits of the mountains. And the water that passes through a pent-up channel within the body of the mountain like a dead thing will not emerge from its first low state, because it is not warmed by the vital heat of the first spring. Moreover the warmth of the element of fire, and by day the heat*

*of the sun, have power to stir up the dampness of the low places and draw this to a height in the same way as it draws the clouds and calls up their moisture from the expanses of the sea.*" [Page 199, Suh, 2005].

Humor is Latin for moisture. da Vinci, who is recognised as one of history's most brilliant minds, lived 500 years after Karaji's time. We may appreciate Karaji's profound knowledge of hydrology, especially when appreciated in the context of his time. da Vinci was clearly on the incorrect path with water flowing uphill, however, Karaji seems to be very close to understanding

the core of hydrologic cycle and the mechanism of water flow from higher ground levels to lower levels. We note that it was just in the seventeenth century that a clear understanding of hydrologic cycle was realized (Todd and Mays, 2004).

Fascinatingly, the protection boundary of wells and qanats based on religious laws are described. For example, Karaji explained that whoever dug a well, with the permission of the ruler, the digger would be the owner of the well. There would also be a protection zone of 40 cubits for this well (about 20 m). However, if the well was established illegally, the digger

would not be the owner and there is no protection zone for that. The protection zone for Qanat is 500 cubits (250 m) [Page 67: Xadiv Jam, 1966]. It is intriguing to note that the only available and ruling law at the time in the Islamic world was strictly based on religious ideas and texts. Thus, all matters relating to ownership, property and rights were based entirely on religious ideas and works. These were developed, promoted, espoused and written entirely by religious scholars. They were linked to the practice of the prophet Muhammad and his companions' practices. Karaji's work began to bring science, engineering,

maths and technology to this important – and at that time entirely religious – legal discussion, principles and practice.

Next, Karaji defined protection limits based on his knowledge and consideration of differing soil types. "*The protection areas of Qanat in hard soils is less than that for loose soils.*" [Translated from Page 74: Xadiv Jam, 1966]. Karaji understands that wells placed in the more permeable material (the loose soils) require a greater area or size for the groundwater protection zone around it compared to that in the less permeable material (the hard soils). Groundwater protection or buffer zones are based

on the very same principle today – a principle that Karaji conceived a thousand years ago. We speculate that what Karaji mentioned here is related to his intuitive understanding of the ease of water flow in loose soils compared to that in hard soils. It is possible that Karaji understood that water flowed more easily through loose soils than it did through hard soils – leading to a concomitant increase in the size of the protection zone for a well in the more permeable material (the loose materials). This may be some of the very earliest documented insights into the rates and ease of groundwater flow through different

geologic materials – earliest conceptions of what we would call hydraulic conductivity today. They are also earliest documented conceptions of modern-day hydrogeology.



Then Karaji reported possible complications during excavation and described the technical solutions to overcome them. Moreover, he reported how to prepare the construction works and prepare aqueducts. He provided detailed methods to level the construction sites and illustrated the apparatus that can be used for levelling in both horizontal and vertical directions and

the methods for surveying and planning Qanat construction [Pages 93-141: Xadiv Jam, 1966]. Figures 3-8 illustrate diagrams and schematics from a later-century copy of the original manuscript of Karaji's book showing surveying and levelling apparatuses, as well as, the proofs and descriptions of their applications (Karaji, 1675). Karaji provided elaborate explanations on stabilizing techniques for tunnel excavation procedures [Pages 142-150: Xadiv Jam, 1966]. He explained how to plan and dig in a tortuous conduct and how to open, maintain, and dredge Qanats [Pages 151-162: Xadiv Jam, 1966]. Figure 9 illustrates

a caliper, a ruler and the schematic for planning how to dig in a tortuous Qanat (Karaji, 1675).

## 3 Epilogue

Karaji's pioneering scientific and engineering contributions into hydrology and hydrogeology through his book "The Extraction of Hidden Waters" are seminal and significant. Despite this, we and other authors have noted that his contributions to hydrologic science have been largely unknown and hence greatly undervalued and underappreciated. The fact that full

translations of his work into other languages did not exist until modern time (e.g., French translation in 1973, Italian in 2007, and English in 2011) is probably a key reason for this. The situation may have been different if translations had occurred much earlier, but this was not common at the time. Thus, his contributions, we surmise, were simply not known.

It is abundantly clear from our article, and a small number of previous papers on this matter, that Karaji both thought about and proposed interesting, important and prescient ideas about hydrology hundreds of years before European thinkers in the

Middle Ages. Many of those have stood the test of time and are as true today as they were a thousand years ago. Karaji was a prognostic hydrologist and hydrogeologist hundreds of years ahead of his time. Beyond the specific topic of Karaji's book on the extraction of hidden waters, the comprehensive content, details and topics that he has covered in the book are very impressive for engineering construction project management. This important point has not been noted before, to the best of our knowledge. Therefore, Karaji's book is not only "the oldest textbook on hydrology", but also the first book on engineering

construction management! Like previous authors, we too assert Karaji deserves more credit in hydrologic science and engineering than has been the case to date. We hope our paper plays a part in rectifying this. We hope that it helps to bring Karaji – the scientist and his science – to the attention of current and future generations of hydrologists around the world.

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






**Figure 1: Abubakr Mohammad Karaji (c. 953 – c. 1029) statue, created by Manouchehr Abollahzadeh, placed at the Water Museum of Sa'dabad Museum Complex in Tehran, Iran (http://sadmu.ir/post/6 ).**





**Figure 2: The first page of Inbāt al-miyāh al-khafiya. Page 1v from Karaji (1675). Permanent Link:**
**http://hdl.library.upenn.edu/1017/d/medren/9948256513503681**






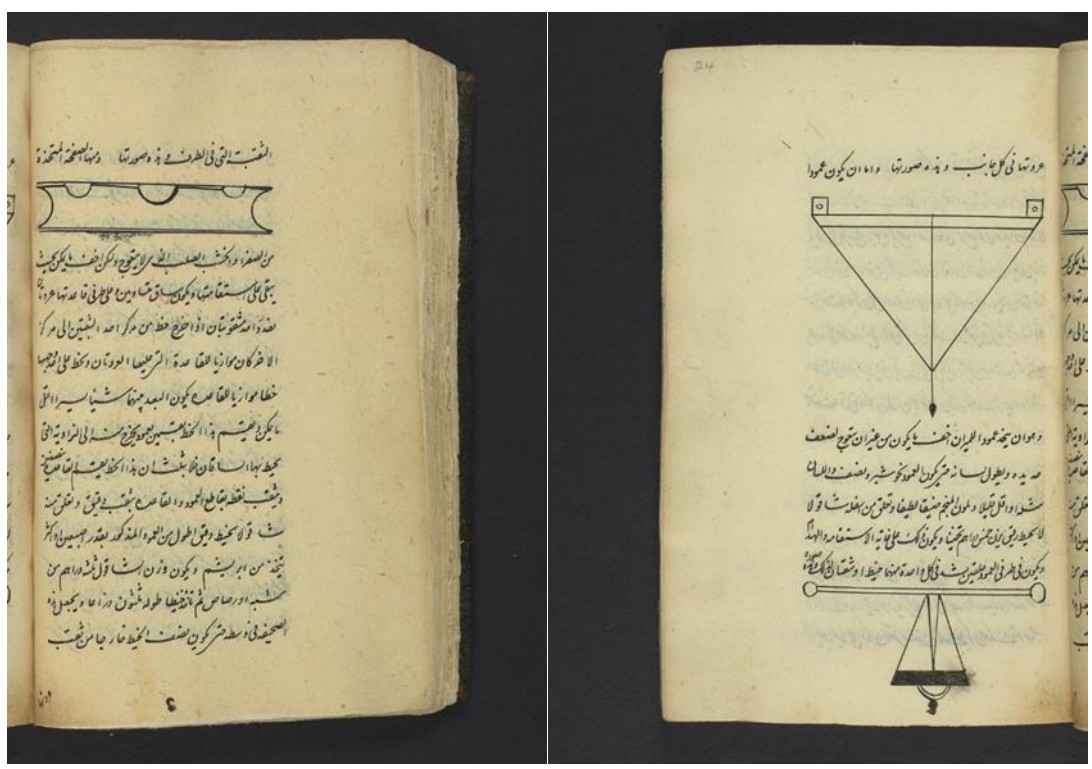

**Figure 3: Illustrations of surveyor's level apparatuses. Diagrams f23v and f24r from Karaji (1675). Permanent Link: http://hdl.library.upenn.edu/1017/d/medren/9948256513503681**





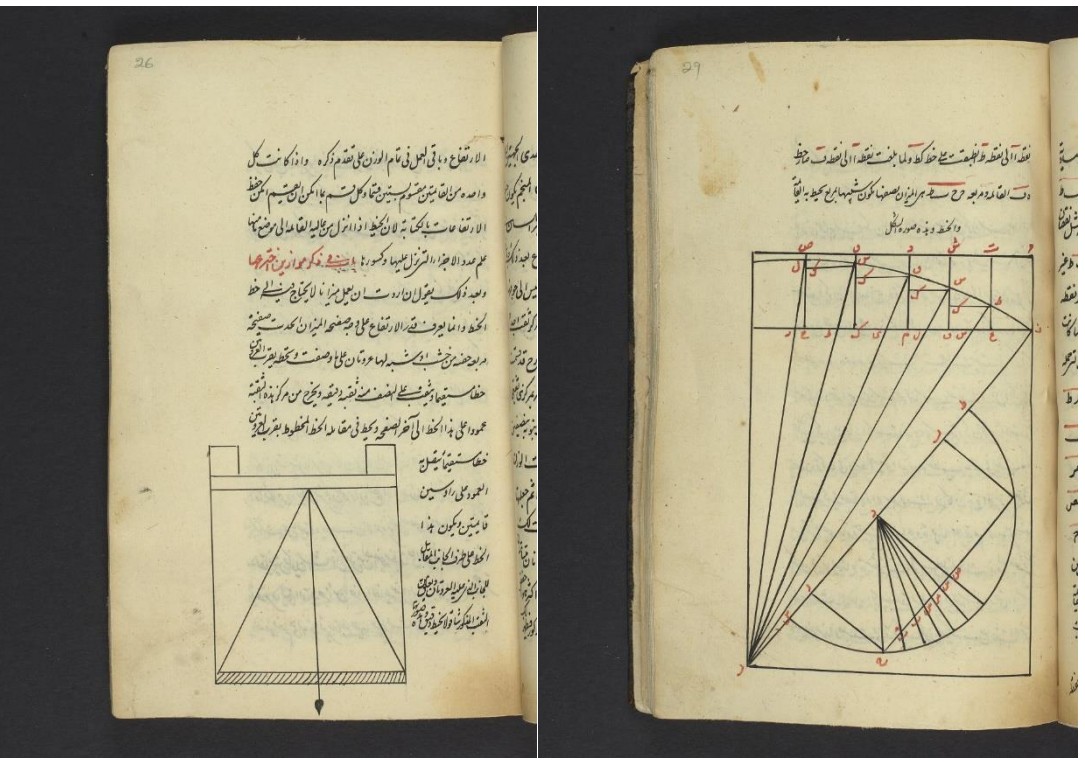

**Figure 4: Illustrations of surveyor's level apparatuse invented by Karaji and the proof and description on its application. Diagrams f26r and f29r from Karaji (1675). Permanent Link: http://hdl.library.upenn.edu/1017/d/medren/9948256513503681**





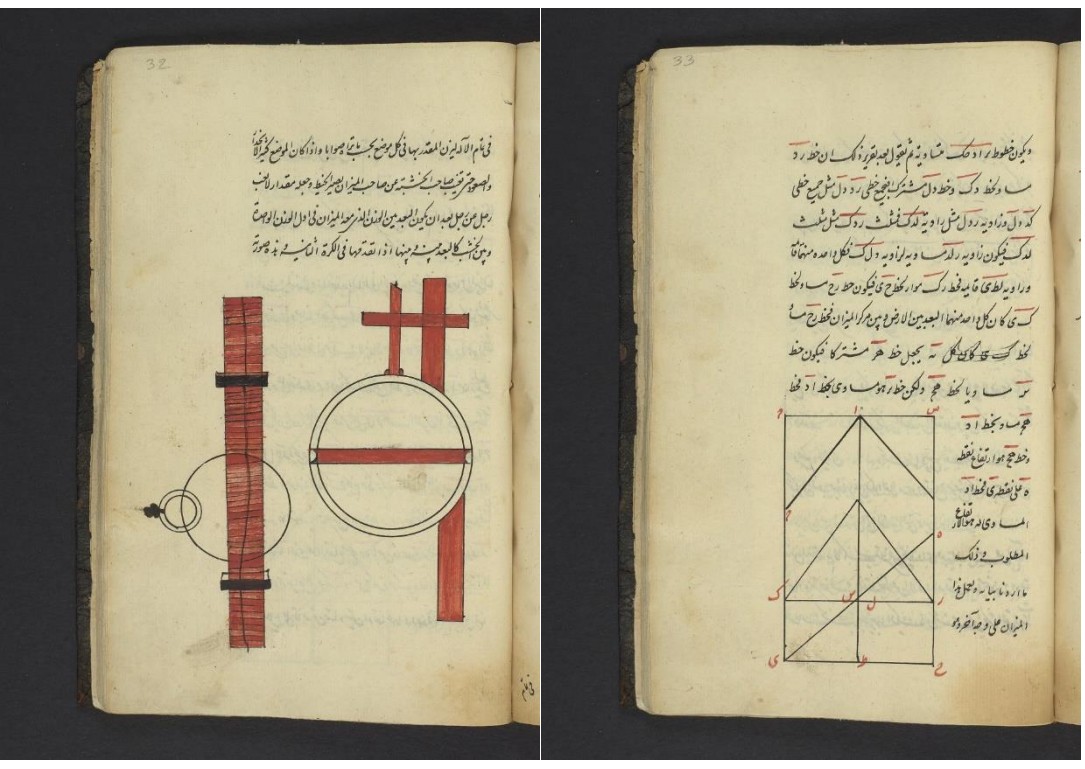

**Figure 5: Illustrations of surveying apparatuses to measure distance and level and the proof and description on its application. Diagrams f32r and f33r from Karaji (1675). Permanent Link: http://hdl.library.upenn.edu/1017/d/medren/9948256513503681**



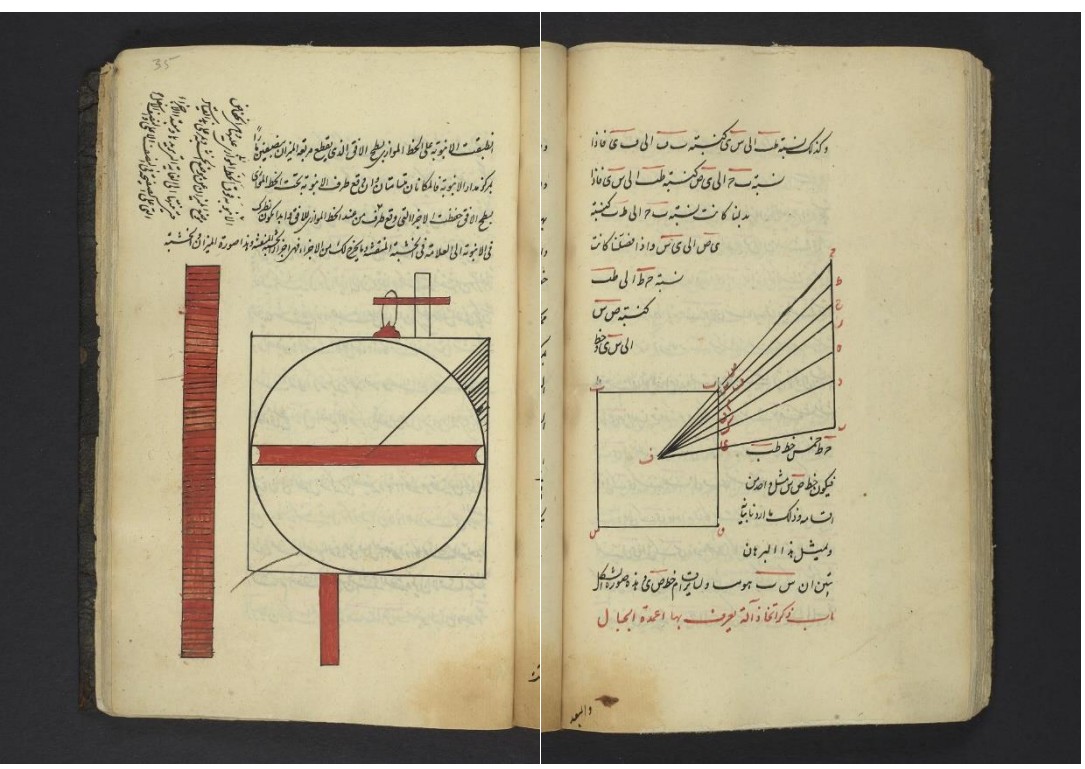


**Figure 6: Illustration for the proof on how to vertically to describe how to measure and determine the height of a mountain. Diagrams f35r and f36v from Karaji (1675). Permanent Link: http://hdl.library.upenn.edu/1017/d/medren/9948256513503681**





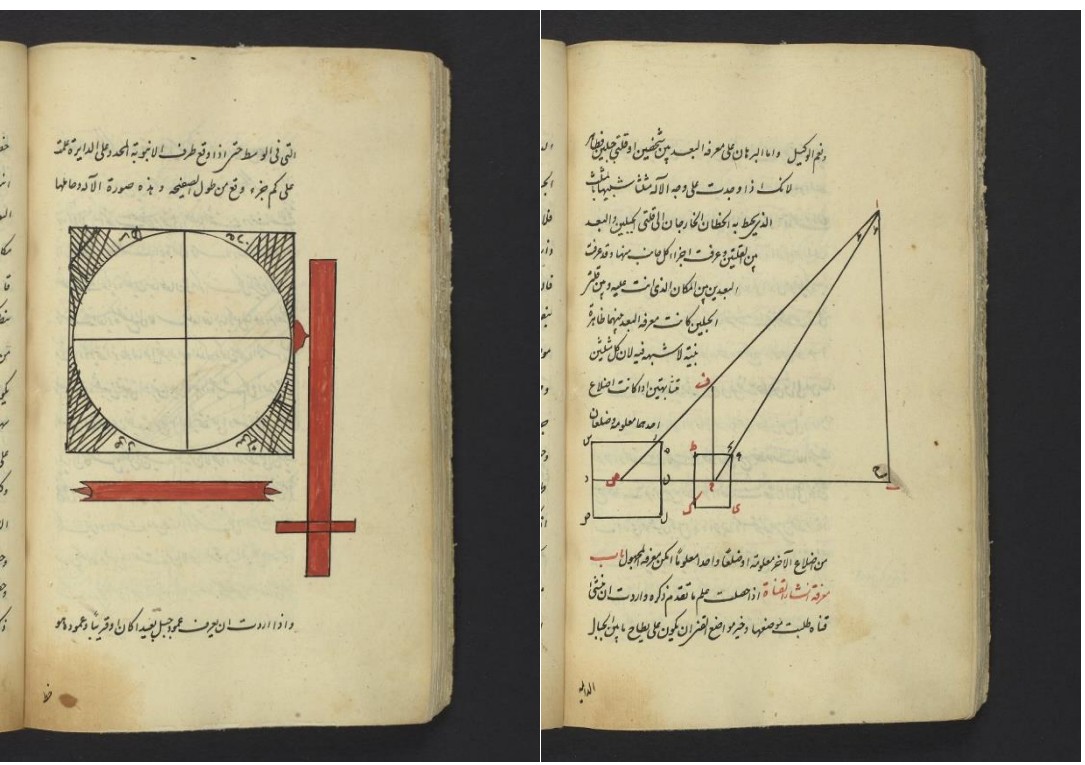

**Figure 7: Illustrations for the apparatus and to describe how to measure and determine the height of a mountain. Diagrams f37v and f40v from Karaji (1675). Permanent Link: http://hdl.library.upenn.edu/1017/d/medren/9948256513503681**





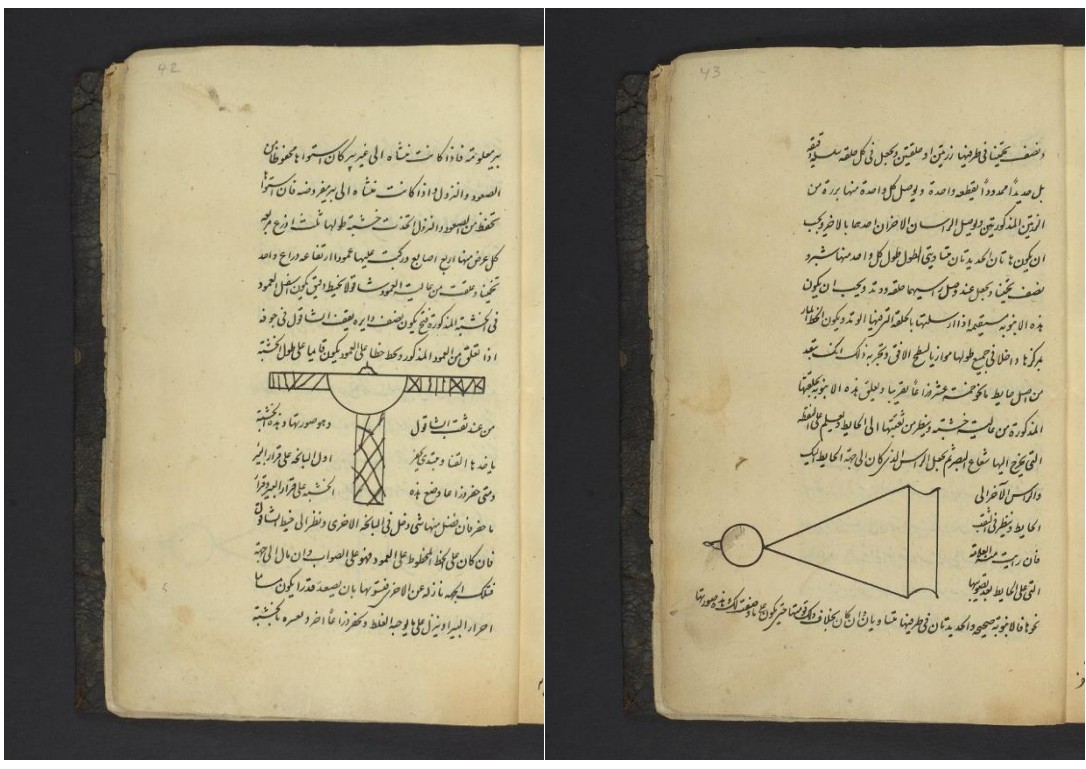

**Figure 8: Illustration of the apparatuses for checking the straightness of the aqueduct. Diagram f42r and f43r from Karaji (1675).
Permanent Link: http://hdl.library.upenn.edu/1017/d/medren/9948256513503681**



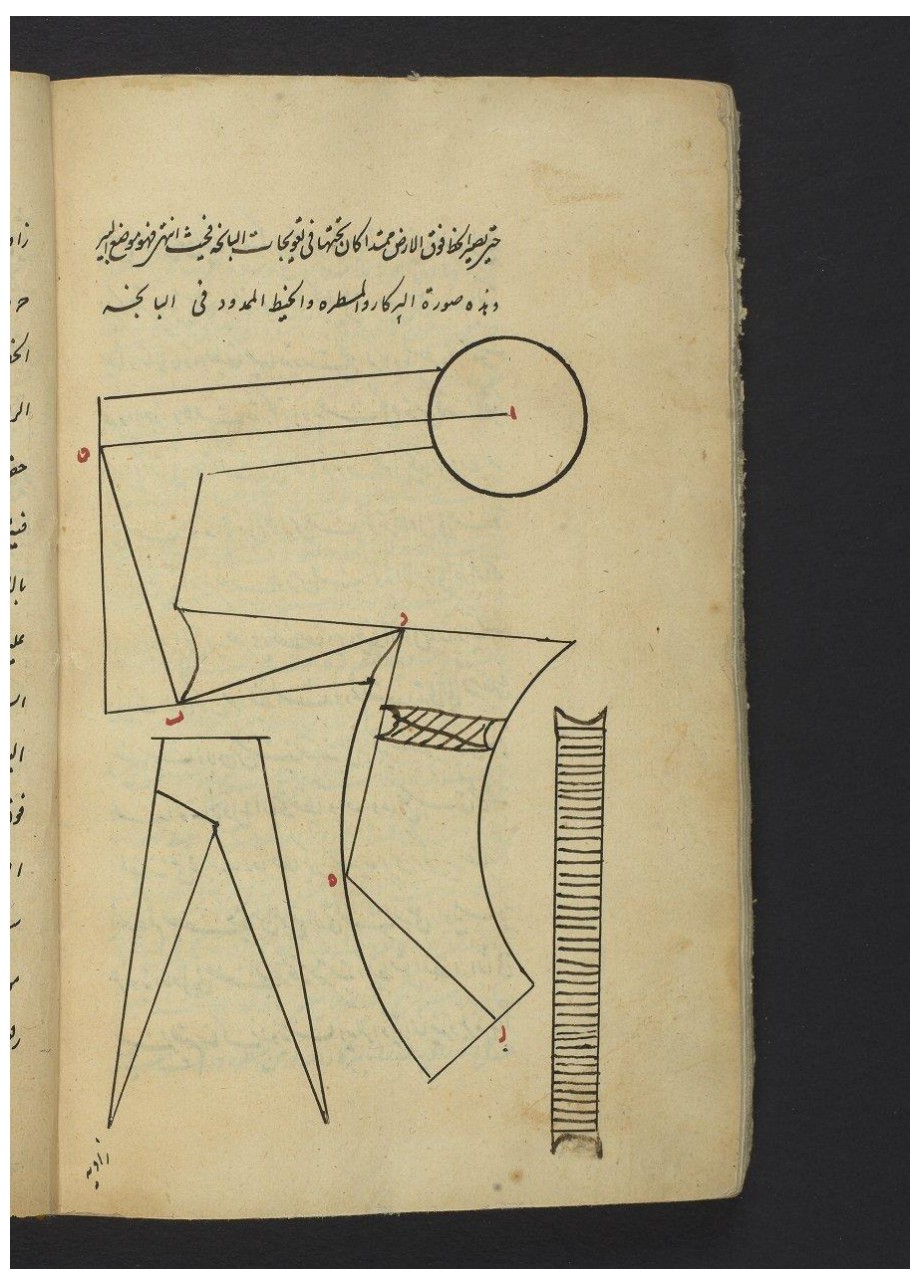

**Figure 9: Illustrations of calliper, ruler and planning how to dig in a tortuous Qanat. Diagram f45v from Karaji (1675). Permanent Link: http://hdl.library.upenn.edu/1017/d/medren/9948256513503681**