# Peer review of "The millennium old hydrogeology textbook "The Extraction of Hidden Waters" by the Persian mathematician and engineer Abubakr Mohammad Karaji (c. 953 – c. 1029)"

_Hydrology and Earth System Sciences, 2019_

## Short Comment (SC1) · 4 Sep 2019

First of all, I think Ataie-Ashtiani and Simmons cover a very exciting and yet overlooked book, that definitely deserves more attention. Historical notes on non-European works that cover hydrology are rare, even though dismissing these leads to a gap in scientific history. Limited scientific progress was made in Europe between the fall of the Roman Empire and the Renaissance, as far as I am aware of. Many ancient civilizations outside Europe existed that carefully managed their water and must have had an understanding of hydrology.

The paper starts with a concise background on Karaji's "The Extraction of Hidden Waters", providing context of Karaji's time and place, other important findings of Karaji, and possible motivations for the book. Following this, several important parts of Karaji's book are summed: The description of Qanats, the pragmatic topics the book covers, followed by a selection of quotes that highlight Karaji's understanding of hydrology and a comparison to what later authors wrote.

I am under the impression that the authors have thoroughly read later European works, but I think they have missed a Roman work. Vitruvius wrote a series of ten books called "de Architectura" (on Architecture). The eighth book of this series covers water, in which he discusses several topics that were also discussed in The Extraction of Hidden Waters. Even though I personally share the authors' excitement on the importance of Karaji's work, I think some more care should be taken with claims that something is the "first" or "earliest".

First of all, it is stated that Karaji's "The Extraction of Hidden Waters" is the oldest textbook on hydrology (line 254). This of course depends on when a textbook covers a topic *x* thoroughly enough to be called "a book on *x*". I think one important criterium should be a discussion of the hydrological cycle, as Karaji did. Vitruvius discusses the hydrological cycle in his chapter "Rainwater":

*"...And rainfall is not abundant in the plains, but rather on the mountains or close to mountains, for the reason that the vapour which is set in motion at sunrise in the morning, leaves the earth, and drives the air before it through the heaven in whatever direction it inclines; then, when once in motion, it has currents of air rushing after it, on account of the void which it leaves behind. ... Wherever the winds carry the vapour which rolls in masses from springs, rivers, marshes, and the sea, it is brought together by the heat of the sun, drawn off, and carried upward in the form of clouds; then these*

*clouds are supported by the current of air until they come to mountains, where they are broken up from the shock of the collision and the gales, turn into water on account of their own fullness and weight, and in that form are dispersed upon the earth."* (Vitr. 8.2.1 and 8.2.2 translated by Milford, 1914).

Futhermore, the authors suggest in lines 229-230 that Karaji's work may show the earliest (written) conception of hydraulic conductivity. An earlier potential of understanding of hydraulic conductivity can be found in Vitruvius:

*"Searchers for water must also study the nature of different localities; for those in which it is found are well defined. In clay the supply is poor, meagre, and at no great depth. It will not have the best taste. In fine gravel the supply is also poor, but it will be found at a greater depth. It will be muddy and not sweet. In black earth some slight drippings and drops are found that gather from the storms of winter and settle down in compact, hard places. They have the best taste. Among pebbles the veins found are moderate, and not to be depended upon. These, too, are extremely sweet. In coarse grained gravel and carbuncular sand the supply is surer and more lasting, and it has a good taste. In red tufa it is copious and good, if it does not run down through the fissures and escape."* (Vitr. 8.1.2 translated by Milford, 1914).

Finally, just like in The Extraction of Hidden Waters, several pragmatic aspects of water management in book 8 of De Architectura. Listing the chapter titles: *How to find water; rainwater; various properties of different waters; tests of good water; levelling and levelling instruments; aqueducts, wells, and cisterns.* Furthermore, book 10 of De Architectura covers the construction and workings of several contraptions that are relevant to water management: *engines for raising water; water wheels and water mills; and the water screw.* Therefore, Vitruvius approaches Karaji in the range of covered topics.

To conclude, I think it is very debatable whether Extraction of Hidden Waters is the first textbook on hydrology. I do however think it can be argued that it is the first known

textbook on hydrogeology. In addition, Karaji seems to cover several topics that are not covered by Vitruvius: the construction of Qanats, maintenance guides, well protection zones, and a better understanding of how water flows through the subsurface. It therefore appears to me that the Extraction of Hidden Waters is a lot more complete than De Architectura book 8.

I encourage the authors to look into De Architectura and briefly compare it with The Extraction of Hidden Waters. An English translation of his book can be found here: http://www.perseus.tufts.edu/hopper/text?doc=Perseus

Furthermore there is also a paper that discusses the ancient Chinese notes on hydrogeology, that might be of interest to you as well:

Zhou, Y., Zwahlen, F. and Wang, Y.: The ancient Chinese notes on hydrogeology, Hydrogeol. J., 19(5), 1103–1114, doi:10.1007/s10040-010-0682-1, 2011.

---

## Referee Comment (RC1) · Stathis C. Stiros (Referee) · 23 Oct 2019

The manuscript by Ataie-Ashtiani and Simmons discuss the novelty and importance of the 11c. book of Al Karaji "Extraction of hidden waters" for the exploitation of subsurface waters in Medieval times using the qanat technology. This is an important, poorly known interdisciplinary topic covering Hydrology, Geotechnical Engineering and Geodesy, with various important implications and suitable for the Special Volume of

HESS on the 'History of Hydrology'. The manuscript, however, has two main problems, which call for major revision, mainly focusing on the need for a more critical and technical approach. A) A first problem is that the manuscript is describing the work and personality of Al Karaji from a rather narrow point of view. His contribution cannot and should not be underestimated, but the context and possible background of his work should be noticed. A1) Some pioneers in the study of qanats such as Wulff and English, cited by the authors, notice a "Book on qanats", written about 100 years before Al Karaji and which covers at least some aspects of Al Karaji's book. A2) Persian engineering during the Abbasid period described by Al Karaji is likely to summarize a knowledge and experience which was both produced in Persia and imported from other regions. Clearly, early engineers in Persia and the surrounding area had developed a technology for the construction of the first qanats in favourable rock conditions (what can be currently classified as soft soils), probably since 1000BC (and not since 3000BC, as marked in line 89), but at the same period there have been impressive engineering works in the regions covered by modern Greece (Mycenean era, before 1000BC) and modern Italy (circa 700BC, Etruscans) (see for example Angelakis et al, 2003). Furthermore, Eupalinus had constructed in Samos Island, Aegean Sea, a 1000m long tunnel from two openings only, with a second, qanat-type tunnel beneath it through both unstable and hard rock (see Kienast, 1995); this testifies to an ancient technology and science which have probably influenced later periods, including Al Karaji (cf. Lewis, 2001). The Persian expansion to Egypt during the Achaemenid period was most probably facilitated by the adaptation of the qanat technology to Egypt, but Persians probably benefited from the knowledge of surface waters by ancient Egyptians who had developed specific metrological techniques ("Nilometers"). Apart from a mutual transfer of technology in hydrological engineering between ancient Persia and adjacent regions, at a later stage, there might have been also a transfer of Roman water technology (for example, Grewe, 1998), summarized by Vitruvius, the work of which was possibly known to Persian intellectuals. B) A second point noticed in Stiros (2006) is that Al Karaji (and all other ancient writers) on one hand was subject to strict

limitations in publicization of critical technical information, which was limited to muqannis, of specific guild-type groups working on qanats till the sub-modern era; this makes ancient books different from modern technical manuals. On the other hand, Al Karaji book reveals that he had the SENSE of engineering (for example concerning his understanding of accuracies) and he was most probably aware of critical details of the construction and exploitation of qanats. In this framework, his book included several figures, in some analogy to the book of Agricola, and this was rare in the ancient world. These figures (which are currently freely accessible) are the most important and less well understood point of his work (only some have been commended by Lewis 2001), and they deserve some explanation. I am afraid that in its present form, the manuscript does not permit to the average reader to understand what these figures mean and the techniques used in antiquity to construct aqueducts. I believe that this problem can be easily overpassed, adding some explanations for each figure. Such explanations need not be very technical and detailed, as for example in Stiros (2012) for the leveling of qanats or in Lewis (2001), but it is enough to add next to each ancient figure an explanatory graph and a short text to summarize their significance. For example, in figure 9 for the alignment of the tunnel, it is suggested to use some shading for the rock, to mark the shaft and the tunnel axis (I guess marked with letters in the original figure), to explain some symbols used and also provide an order of magnitude of the scale indicated. For some figures, the comments of Lewis 2001 on Al Karaji (especially pages 298-302) will be very helpful. These changes will lead to a useful and well documented article, suitable for the Special Issue of HESS on the History of Hydrology. S. Stiros

References Angelakis, N. et al., 2013. Minoan and Etruscan Hydro-Technologies. Water, 5(3), 972-987; doi: 10.3390/w5030972 Grewe, K., 1998. Licht am Ende des Tunnels. Plannung und Trassierung im antiken Tunnlebau. Verlag Philip von Zabern, Mainz am Rhein: 218. Kienast, H., 1995. Die Wasserleitung des Eupalinos auf Samos. Deutsches Archaeologisches Institut, Samos Band XIX. pp. 229, pls. 41, figs. 58, foldout plans 3, tables 5. Rudolf Habelt, Bonn. Lewis, M., 2001. Surveying Instruments of Greece and Rome, Cambridge Univ Press Stiros, S., 2006. Accurate measurements

with primitive instruments: The "paradox" in the qanat design, Journal of Archaeological Science, 33, 1058-1064, doi: 10.1016/j.jas.2005.11.013 Stiros, S., 2012. Levelling in antiquity: instrumentation, techniques and accuracies, Survey Review, 44 (32), 45-52, doi: 10.1179/1752270611Y.0000000008

---

## Referee Comment (RC2) · Anonymous Referee #2 · 1 Nov 2019

L21-22: the texts ØğØÍÙĹØÍÚ'Øś ÙĚØ▪ÙĚØŕ ØÍÙE ØğÙĐØ▪ØşÙE ØğÙĐÚ'ØśØňÛŇ-ØğÙEØÍØğØů ØğÙĐÙĚÛŇØğÙĞ ØğÙĐØőÙ ÂÛŇÙĞ are Arabic and not Persian L54-55: the phrase "Instead of Europe" is redundant. I suggest to neglect this phrase. L74: what is the reference for this sentence" Karaji returned to his homeland wrote the book". Please mention the evidence related to this allegation. L254: The authors are requested to develop the idea that claims "the book is the first book on engineering construction management". It needs more proofs and analysis

---

## Author Comment (AC1) · 18 Nov 2019

By: Joeri van Engelen
First of all, I think Ataie-Ashtiani and Simmons cover a very exciting and yet overlooked book, that definitely deserves more attention. Historical notes on non-European works that cover hydrology are rare, even though dismissing these leads to a gap in scientific history. Limited scientific progress was made in Europe between the fall of the Roman Empire and the Renaissance, as far as I am aware of. Many ancient civilizations outside Europe existed that carefully managed their water and must have had an understanding of hydrology. The paper starts with a concise background on Karaji's "The Extraction of Hidden Waters", providing context of Karaji's time and place, other important findings of Karaji, and possible motivations for the book. Following this, several important parts of Karaji's book are summed: The description of Qanats, the pragmatic topics the book covers, followed by a selection of quotes that highlight Karaji's understanding of hydrology and a comparison to what later authors wrote.

Response:

We appreciate the positive appraisal of the commentator and the useful comments that will be addressed in the following response.

Comment:

I am under the impression that the authors have thoroughly read later European works, but I think they have missed a Roman work. Vitruvius wrote a series of ten books called "de Architectura" (on Architecture). The eighth book of this series covers water, in which he discusses several topics that were also discussed in The Extraction of Hidden Waters. Even though I personally share the authors' excitement on the importance of Karaji's work, I think some more care should be taken with claims that something is the "first" or "earliest". First of all, it is stated that Karaji's "The Extraction of Hidden Waters" is the oldest textbook on hydrology (line 254). This of course depends on when a textbook covers a topic x thoroughly enough to be called "a book on x". I think one important criterium should be a discussion of the hydrological cycle, as Karaji did.

[Figure]

Vitruvius discusses the hydrological cycle in his chapter "Rainwater": "…And rainfall is not abundant in the plains, but rather on the mountains or close to mountains, for the reason that the vapour which is set in motion at sunrise in the morning, leaves the earth, and drives the air before it through the heaven in whatever direction it inclines; then, when once in motion, it has currents of air rushing after it, on account of the void which it leaves behind. … Wherever the winds carry the vapour which rolls in masses from springs, rivers, marshes, and the sea, it is brought together by the heat of the sun, drawn off, and carried upward in the form of clouds; then these clouds are supported by the current of air until they come to mountains, where they are broken up from the shock of the collision and the gales, turn into water on account of their own fullness and weight, and in that form are dispersed upon the earth." (Vitr. 8.2.1 and 8.2.2 translated by Milford, 1914). Response: We would like to emphasize that the objective of this essay was to celebrate Karaji's book and contributions, without any attempt to downgrade any other possible contributions from others. It is not a detailed comparative analysis. For example, in L16 we mentioned: "Although some of the ideas may have been presented elsewhere,...". Therefore, we do not claim that all the ideas I this book were presented for the first time by Karaji. We have claimed that "L16:…, this is the first time that a whole book was focused on different aspects of hydrology and hydrogeology.". To the best of our knowledge this is a valid claim. In Vitruvius's de Architectura or The Ten Books on Architecture, although there are chapters/books (in particular Book VIII-pages 225 to 251 in Morgan's translation) regarding water and hydrology, the de Architectura is essentially about Architecture. We understand the commenter's point on the use of the "first". We have modified it to " …, to the best of our knowledge, this is the first time that a whole book was focused on different aspects of hydrology and hydrogeology". We have relaxed our assertion about the book being "the first" of its kind. We have changed "the book is the first book on …" to "the book is among the earliest known texts on … ". Futhermore, the authors suggest in lines 229-230 that Karaji's work may show the earliest (written) conception of hydraulic conductivity. An earlier potential of understanding of hydraulic conductivity

can be found in Vitruvius: "Searchers for water must also study the nature of different localities; for those in which it is found are well defined. In clay the supply is poor, meagre, and at no great depth. It will not have the best taste. In fine gravel the supply is also poor, but it will be found at a greater depth. It will be muddy and not sweet. In black earth some slight drippings and drops are found that gather from the storms of winter and settle down in compact, hard places. They have the best taste. Among pebbles the veins found are moderate, and not to be depended upon. These, too, are extremely sweet. In coarse grained gravel and carbuncular sand the supply is surer and more lasting, and it has a good taste. In red tufa it is copious and good, if it does not run down through the fissures and escape." (Vitr. 8.1.2 translated by Milford, 1914).

Response:

As seen, our conjecture about the Karaji's earliest conceptions of hydraulic conductivity based on his conception of protection area. In L219-231. We wrote: "Next, Karaji defined protection limits based on his knowledge and consideration of differing soil types. "The protection areas of Qanat in hard soils is less than that for loose soils." [Translated from Page 74: Xadiv Jam, 1966]. Karaji understands that wells placed in the more permeable material (the loose soils) require a greater area or size for the groundwater protection zone around it compared to that in the less permeable material (the hard soils). Groundwater protection or buffer zones are based on the very same principle today – a principle that Karaji conceived a thousand years ago. We speculate that what Karaji mentioned here is related to his intuitive understanding of the ease of water flow in loose soils compared to that in hard soils. It is possible that Karaji understood that water flowed more easily through loose soils than it did through hard soils – leading to a concomitant increase in the size of the protection zone for a well in the more permeable material (the loose materials). This may be some of the very earliest documented insights into the rates and ease of groundwater flow through different geologic materials – earliest conceptions of what we would call hydraulic conductivity today. They are also among the earliest known documented

conceptions of modern-day hydrogeology."

Comment:

Finally, just like in The Extraction of Hidden Waters, several pragmatic aspects of water management in book 8 of De Architectura. Listing the chapter titles: How to find water; rainwater; various properties of different waters; tests of good water; levelling and levelling instruments; aqueducts, wells, and cisterns. Furthermore, book 10 of De Architectura covers the construction and workings of several contraptions that are relevant to water management: engines for raising water; water wheels and water mills; and the water screw. Therefore, Vitruvius approaches Karaji in the range of covered topics. To conclude, I think it is very debatable whether Extraction of Hidden Waters is the first textbook on hydrology. I do however think it can be argued that it is the first known textbook on hydrogeology. In addition, Karaji seems to cover several topics that are not covered by Vitruvius: the construction of Qanats, maintenance guides, well protection zones, and a better understanding of how water flows through the subsurface. It therefore appears to me that the Extraction of Hidden Waters is a lot more complete than De Architectura book 8. I encourage the authors to look into De Architectura and briefly compare it with The Extraction of Hidden Waters. An English translation of his book can be found here: http://www.perseus.tufts.edu/hopper/text?doc=Perseus Furthermore there is also a paper that discusses the ancient Chinese notes on hydrogeology, that might be of interest to you as well: Zhou, Y., Zwahlen, F. and Wang, Y.: The ancient Chinese notes on hydrogeology, Hydrogeol. J., 19(5), 1103–1114, doi:10.1007/s10040-010-0682-1, 2011.

Response:

To cover a complete historical description of the development of hydrology and to consider the contributions of different ancient states such as Chinese, Greeks, Persians (and others), and is beyond the objective and scope of this essay. We could not do this synthesis and comparative analysis justice in a brief summary. As mentioned, we are

not aware of any other book before Karaji's book that was a whole book focused on different aspects of hydrology and hydrogeology. We agree with the commentator that it can be argued that it is the first known textbook on hydrogeology. As noted earlier, we have modified the text to relax the notion of the first textbook to ". . ., to the best of our knowledge, this is the first time that a whole book was focused on different aspects of hydrology and hydrogeology".

Please also note the supplement to this comment:
https://www.hydrol-earth-syst-sci-discuss.net/hess-2019-407/hess-2019-407-AC1-supplement.pdf

---

## Author Comment (AC2) · 18 Nov 2019

By: Stathis C. Stiros (Referee)
The manuscript by Ataie-Ashtiani and Simmons discuss the novelty and importance of the 11c. book of Al Karaji "Extraction of hidden waters" for the exploitation of subsurface waters in Medieval times using the qanat technology. This is an important, poorly known interdisciplinary topic covering Hydrology, Geotechnical Engineering and Geodesy, with various important implications and suitable for the Special Volume of HESS on the 'History of Hydrology'.

Response:

We appreciate the positive appraisal by Professor Stiros and the detailed and helpful comments that will be addressed in the following response.

Comment:

The manuscript, however, has two main problems, which call for major revision, mainly focusing on the need for a more critical and technical approach. A) A first problem is that the manuscript is describing the work and personality of Al Karaji from a rather narrow point of view. His contribution cannot and should not be underestimated, but the context and possible background of his work should be noticed. A1) Some pioneers in the study of qanats such as Wulff and English, cited by the authors, notice a "Book on qanats", written about 100 years before Al Karaji and which covers at least some aspects of Al Karaji's book. A2) Persian engineering during the Abbasid period described by Al Karaji is likely to summarize a knowledge and experience which was both produced in Persia and imported from other regions. Clearly, early engineers in Persia and the surrounding area had developed a technology for the construction of the first qanats in favourable rock conditions (what can be currently classified as soft soils), probably since 1000BC (and not since 3000BC, as marked in line 89), but at the same period there have been impressive engineering works in the regions covered by modern Greece (Mycenean era, before 1000BC) and modern Italy (circa 700BC, Etruscans) (see for example Angelakis et al, 2003). Furthermore, Eupalinus had constructed in Samos Island, Aegean Sea, a 1000m long tunnel from two openings only, with a second, qanat-type tunnel beneath it through both unstable and hard rock (see Kienast, 1995); this testifies to an ancient technology and science which have probably influenced later periods, including Al Karaji (cf. Lewis, 2001). The Persian expansion to Egypt during the Achaemenid period was most probably facilitated by the adaptation of the qanat technology to Egypt, but Persians probably benefited from the knowledge of surface waters by ancient Egyptians who had developed specific metrological techniques ("Nilometers"). Apart from a mutual transfer of technology in hydrological engineering between ancient Persia and adjacent regions, at a later stage, there might have been also a transfer of Roman water technology (for example, Grewe, 1998), summarized by Vitruvius, the work of which was possibly known to Persian intellectuals.

Response:

We agree with the reviewer's comment. We had tried to implicitly highlight that Karaji was standing on the foundations of knowledge that may have been laid down by the people who lived before him. For example, in L16 we mentioned: "Although some of the ideas may have been presented elsewhere,..." and in L49-55 we wrote: " Karaji lived in Baghdad under the Abbasid rulers. We anticipate that he would have been a direct beneficiary of the translation movement. This initiative was begun under the second Caliph Al-Mansur and continuing through to the seventh Caliph Al-Ma'mun and saw a large amount of significant scientific, religious and other literature translated into Arabic for scholars to use. At this time, Baghdad was one of the world's greatest places of learning and knowledge. It hosted some of the world's best libraries. It was a vibrant place for scholarly activity and scientific discovery. The Middle East became the centre of intellectual thought instead of Europe." The focus and purpose of this essay was on Karaji's book and contributions, without any attempt to downgrade any other possible contributions from others. It is not a comparative analysis. L34-37: "We believe that Karaji's contributions in hydrology and hydrogeology are significant and

should be remembered and revisited in this Hydrology and Earth System Sciences special issue on the 'History of Hydrology'. In this essay, we revisit this book and provide an English translation of the pieces from the book that crucially offer pioneering ideas in hydrogeology and in general for engineering projects". Therefore, it is beyond the scope of this easy to provide an exact historical audit for the contributions of ancient Greeks, Chinese, Indians, and Persians and others in Hydrology and hydrogeology.

Comment:

B) A second point noticed in Stiros (2006) is that Al Karaji (and all other ancient writers) on one hand was subject to strict limitations in publicization of critical technical information, which was limited to muqannis, of specific guild-type groups working on qanats till the sub-modern era; this makes ancient books different from modern technical manuals. On the other hand, Al Karaji book reveals that he had the SENSE of engineering (for example concerning his understanding of accuracies) and he was most probably aware of critical details of the construction and exploitation of qanats. In this framework, his book included several figures, in some analogy to the book of Agricola, and this was rare in the ancient world. These figures (which are currently freely accessible) are the most important and less well understood point of his work (only some have been commended by Lewis 2001), and they deserve some explanation. I am afraid that in its present form, the manuscript does not permit to the average reader to understand what these figures mean and the techniques used in antiquity to construct aqueducts. I believe that this problem can be easily overpassed, adding some explanations for each figure. Such explanations need not be very technical and detailed, as for example in Stiros (2012) for the leveling of qanats or in Lewis (2001), but it is enough to add next to each ancient figure an explanatory graph and a short text to summarize their significance. For example, in figure 9 for the alignment of the tunnel, it is suggested to use some shading for the rock, to mark the shaft and the tunnel axis (I guess marked with letters in the original figure), to explain some symbols used and also provide an order of magnitude of the scale indicated. For some figures, the comments of Lewis 2001

on Al Karaji (especially pages 298-302) will be very helpful. These changes will lead to a useful and well documented article, suitable for the Special Issue of HESS on the History of Hydrology.

Response:

This is animportant observation by Prof. Stiros. We have also mentioned atL112-116: "The titles of the book sections provide a fascinating insight into the wide range of topics that were covered in the book. It is amazing that the book not only covers the conceptual and technical aspects as well as construction guides, it also provides guidelines for maintenance and even advice on how to deliver and consign the project when the development and construction is over. It even touches on important social aspects such as religious regulations. The book is like a construction and maintenance manual for a modern engineering project!" and "L251-254: Beyond the specific topic of Karaji's book on the extraction of hidden waters, the comprehensive content, details and topics that he has covered in the book are very impressive for engineering construction project management. This important point has not been noted before, to the best of our knowledge." Based on the reviewer's comment we have added the following in the revised version of this essay: "Lewis (2001), who explored the history of surveying instruments of Greece and Romans, referred to Karaji's book and his contributions in procedures and inventive instruments for levelling and sighting in surveying engineering. Karaji's ideas in surveying revealed his sense of engineering concerning an understanding of accuracies and awareness of essential elements of the construction and exploitation of qanats (Stiros, 2006)." Also, further explanations have been added in the figure captions to the extent that it is relevant and within the scope of this essay.

Please also note the supplement to this comment:
https://www.hydrol-earth-syst-sci-discuss.net/hess-2019-407/hess-2019-407-AC2-supplement.pdf

---

## Author Comment (AC3) · 18 Nov 2019

Interactive comment on "The millennium old hydrogeology textbook "The Extraction of Hidden Waters" by the Persian mathematician and engineer Abubakr Mohammad Karaji (c. 953–c. 1029)" by Behzad Ataie-Ashtiani and Craig T. Simmons

[Figure]

Comment:

L21-22: the texts (ØğÙĘØÍØğØů ØğÙĎÙĚÛŇØğØą ØğÙĎØőÙĄÛŇÙĞ‎) and (ØğØÍÙĹØÍÚÍØś ÙĚØ∎ÙĚØŕ ØÍÙĘ ØğÙĎØ∎ØşÙĘ ØğÙĎÚÍØśØňÛŇ‎) are Arabic and not Persian

Response:

The reviewer's comment is correct and has been changed in the revised version of the manuscript. As Karaji was originally Persian (see Girogio Levi Della Vida, 1934) and that the Arabic and Persian alphabet are similar, we had used the term Persian in the early version. This has now been corrected.

Comment:

L54-55: the phrase "Instead of Europe" is redundant. I suggest to neglect this phrase.

Response:

Agreed and modified.

Comment:

L74: what is the reference for this sentence" Karaji returned to his homeland wrote the book". Please mention the evidence related to this allegation.

Response:

Karaji is considered to be born in Persia (e.g., Girogio Levi Della Vida, 1934; Xadiv Jam, 1966; Nadji and Voigt, 1972). His return to Persia when he wrote the book has been mentioned in Nadji and Voigt (1972) and Lewis (2001). To the best of our knowledge, this is correct. These references have been added to the modified version.

Comment:

L254: The authors are requested to develop the idea that claims "the book is the first book on engineering construction management". It needs more proofs and analysis

Response:

As it was mentioned in the manuscript this is a speculation by authors as L112-116: "The titles of the book sections provide a fascinating insight into the wide range of topics that were covered in the book. It is amazing that the book not only covers the conceptual and technical aspects as well as construction guides, it also provides guidelines for maintenance and even advice on how to deliver and consign the project when the development and construction is over. It even touches on important social aspects such as religious regulations. The book is like a construction and maintenance manual for a modern engineering project!" and L251-254: "Beyond the specific topic of Karaji's book on the extraction of hidden waters, the comprehensive content, details and topics that he has covered in the book are very impressive for engineering construction project management. This important point has not been noted before, to the best of our knowledge."

We have relaxed our assertion about the book being "the first" of its kind. We have changed "the book is the first book on engineering construction management" to "the book is among the earliest known texts on engineering construction management".

Please also note the supplement to this comment:
https://www.hydrol-earth-syst-sci-discuss.net/hess-2019-407/hess-2019-407-AC3-supplement.pdf

---

## Referee Comment (RC3) · S. Majid Hassanizadeh (Referee) · 19 Nov 2019

Interactive comments on "The millennium old hydrogeology textbook "The Extraction of Hidden Waters" by the Persian mathematician and engineer Abubakr Mohammad Karaji (c. 953–c. 1029)" by Behzad Ataie-Ashtiani and Craig T. Simmons S. Majid Hassanizadeh (Referee) I enjoyed reading this manuscript. It gives a thorough description of Karaji's work, its scientific as well as historical significance, and its practical value. It is definitely a valuable addition to this issue of HESS. I believe the manuscript needs to be improved as some statements are not accurate (please see below for examples). Also, the text needs improvement; I have provided quite a few suggestions in the annotated pdf file. - I would like to suggest including a figure showing a sketch of a qanat and its various elements. This will be beneficial for the reader, and will it easier for the author when explain a qanat (in subsection 1.2, lines 83-90). - The description of figures provided at the end of the manuscript is somewhat superficial and does not really help the reader to understand the figures and their importance. It is also not possible for most readers to read the Arabic text accompanying figures. Aren't these pages translated into English by Schade? If they are, I suggest the authors provide give copies of pages from Schade's book instead of the original Arabic pages. - Qanats were not in use only in arid areas of the Iran, as suggested in line 77. They were in use everywhere in Iran, including mountainous regions in northern (except for the Caspian Sea coast) and western parts of Iran, with plenty of water. - In referring to the qanat tunnel, various words have been used (aqueduct, channel, tunnel) without being clear to the reader that they are all the same thing. I suggest using one word (e.g., tunnel) in all cases. In particular, I suggest avoiding the use of aqueduct, as it is too closely associated with the Roman aqueducts. - Actually, the qanat technology went to Northern Africa before going to Spain. In other words, one could say: "A second major diffusion of Qanat technology occurred with the conquests of Islam into Northern Africa, the peninsular Spain, and the Canary Islands." Also, it is worth mentioning that qanats are found in India as Southerly as Kerala and in Chinese Turkmenistan. - I am not sure the procedure described in lines 173-178 has been really an effective way of water filtration (as suggested in line 177). For water to lose its salinity and heaviness, due to passage through neat ground soil [Isn't this double? ground soil? Why not just soil?], an ion exchange process must occur. So, it must be a soil with some special characteristics. Also, I can't see how water would lose a portion of its salinity and heaviness when leaking from a new pot! Perhaps the authors should elaborate on the potential of this procedure for reducing salinity. - I do not think the presentation in lines

212-220, linking the laws about water rights and safe distances between wells and qanats to Islamic laws, the script, and the prophet Muhammad's practice, is justified. There existed wells and qanats in Iran before Islam and cities and villages had laws and customs ruling such things. Also, I wonder whether exact numbers given by Karaji (lines 214 and 215) can be found in Islamic records. Moreover, I don't see the value of linking Karaj's writings to Islamic laws. If this is needed, I think a more detailed investigation with references, in order to document such a link, should be provided. - Protection zone of wells and qanats is a term used in relation to contamination (i.e. protection from pollution) and not to the use and extraction of water (which is the context in line 212). I think the proper terms here are ownership limits and well boundaries.

Please also note the supplement to this comment:
https://www.hydrol-earth-syst-sci-discuss.net/hess-2019-407/hess-2019-407-RC3-supplement.pdf

**Supplement:**

[Figure]

**The millennium old hydrogeology textbook "The Extraction of Hidden Waters" by the Persian mathematician and engineer Abubakr Mohammad Karaji (c. 953 – c. 1029)**

Behzad Ataie-Ashtiani[1,2], Craig T. Simmons[1]

[1] National Centre for Groundwater Research and Training and College of Science & Engineering, Flinders University, Adelaide, South Australia, Australia
[2] Department of Civil Engineering, Sharif University of Technology, Tehran, Iran

*Correspondence to*: Behzad Ataie-Ashtiani (behzad.ataieashtiani@flinders.edu.au)

**Abstract.** We revisit and shed light on the millennium old hydrogeology textbook "The Extraction of Hidden Waters" by the Persian mathematician and engineer Karaji. Despite the incomparable understanding and conceptualization of the world by the people of that time, ground-breaking ideas and descriptions of hydrological and hydrogeological perceptions such as components of hydrological cycle, groundwater quality and even driving factors of groundwater flow were presented in the book. Although some of the ideas may have been presented elsewhere, this is the first time that a whole book was focused on different aspects of hydrology and hydrogeology. More importantly, we are impressed as the book is composed in a way that covered all aspects that are related to an engineering project including technical and construction issues, guidelines for maintenance, and final delivery of the project when the development and construction is over. We speculate that Karaji's book is the first of its kind to provide a construction and maintenance manual for a modern [I am not sure what is meant by "modern" here. It can't be modern like in the 21st century it has the character of such a manual. Perhaps you mean modern for its time. I am not sure this adjective adds much here.] engineering project.

**1 Prologue**

The eleventh century Arabic book "Inbat al-miyah al-khafiya" (Persian: انباط المیاء الخفیه) [This is acutally not Persian trasnation of the title but its original Arabic title. The same holds for the name of the author; it is written in Arabic] 
[revised manuscript text omitted]

---

## Author Comment (AC4) · 29 Nov 2019

S. Majid Hassanizadeh (Referee) Interactive comment on "The millennium old hydro-geology textbook "The Extraction of Hidden Waters" by the Persian mathematician and engineer Abubakr Mohammad Karaji (c. 953–c. 1029)" by Behzad Ataie-Ashtiani and Craig T. Simmons

Comment:

[Figure]

I enjoyed reading this manuscript. It gives a thorough description of Karaji's work, its scientific as well as historical significance, and its practical value. It is definitely a valuable addition to this issue of HESS. I believe the manuscript needs to be improved as some statements are not accurate (please see below for examples).

Response:

We appreciate the positive appraisal of Prof Hassanizadeh and the useful comments that will be addressed in the following response.

Comment:

Also, the text needs improvement; I have provided quite a few suggestions in the annotated pdf file. - I would like to suggest including a figure showing a sketch of a qanat and its various elements. This will be beneficial for the reader, and will it easier for the author when explain a qanat (in subsection 1.2, lines 83-90).

Response:

With thanks we have considered all the annotated comments on the Pdf file and have mostly implemented them. We have avoided providing a sketch of qanat as we speculate it is known to HESS readers and it can be found easily by any internet search.

Comment:

The description of figures provided at the end of the manuscript is somewhat superficial and does not really help the reader to understand the figures and their importance. It is also not possible for most readers to read the Arabic text accompanying figures. Aren't these pages translated into English by Schade? If they are, I suggest the authors provide give copies of pages from Schade's book instead of the original Arabic pages.

Response:

As this is essay is an historical account of the Karaji's contributions, we suppose the original figures and pictures of the original book would be noteworthy for the readers.

We have provided the essence of the relevant text regarding the figures in the captions to the extent that is matched to the scope and aim of the essay.

Comment:

Qanats were not in use only in arid areas of the Iran, as suggested in line 77. They were in use everywhere in Iran, including mountainous regions in northern (except for the Caspian Sea coast) and western parts of Iran, with plenty of water.

Response:

We have not stated that qanats were only used in the arid areas. The text is "we may speculate that the topic was of great practical interest in the arid area of the Persia plateau." Although there are qanats in the mountainous regions (e.g. in Tehran), the major development and application of this system is in arid and semi-arid regions.

Comment:

In referring to the qanat tunnel, various words have been used (aqueduct, channel, tunnel) without being clear to the reader that they are all the same thing. I suggest using one word (e.g., tunnel) in all cases. In particular, I suggest avoiding the use of aqueduct, as it is too closely associated with the Roman aqueducts.

Response:

Agreed and modified.

Comment:

Actually, the qanat technology went to Northern Africa before going to Spain. In other words, one could say: "A second major diffusion of Qanat technology occurred with the conquests of Islam into Northern Africa, the peninsular Spain, and the Canary Islands." Also, it is worth mentioning that qanats are found in India as Southerly as Kerala and in Chinese Turkmenistan.

Response:

Agreed and modified.

I am not sure the procedure described in lines 173-178 has been really an effective way of water filtration (as suggested in line 177). For water to lose its salinity and heaviness, due to passage through neat ground soil [Isn't this double? ground soil? Why not just soil?], an ion exchange process must occur. So, it must be a soil with some special characteristics. Also, I can't see how water would lose a portion of its salinity and heaviness when leaking from a new pot! Perhaps the authors should elaborate on the potential of this procedure for reducing salinity.

Response:

The major part of the provided lines, the quoted part in italics, are the translations from Karaji's book. In the last sentences we have emphasized that this based on the available knowledge and apparatus of Karai's time: "The treatment Karaji outlined is essentially a water filtration process based on the knowledge and apparatus of the time."

Comment:

I do not think the presentation in lines 212-220, linking the laws about water rights and safe distances between wells and qanats to Islamic laws, the script, and the prophet Muhammad's practice, is justified. There existed wells and qanats in Iran before Islam and cities and villages had laws and customs ruling such things. Also, I wonder whether exact numbers given by Karaji (lines 214 and 215) can be found in Islamic records. Moreover, I don't see the value of linking Karaj's writings to Islamic laws. If this is needed, I think a more detailed investigation with references, in order to document such a link, should be provided.

Response:

The issue protection boundary of wells and qanats based on religious laws was explained by Karaji in his book from page 67 to 79 (Xadiv Jam, 1966). In his explanations he referred to the opinions of Islamic law scholars' (e.g, Hassan Basri, Abu Yousef, Abu Hanifeh) who had referred to prophet Muhammad's practices and sayings. We have added this explanation to the revised manuscript.

Comment:

Protection zone of wells and qanats is a term used in relation to contamination (i.e. protection from pollution) and not to the use and extraction of water (which is the context in line 212). I think the proper terms here are ownership limits and well boundaries.

Response:

Protection zone has been used here in a more general sense that the existing wells and qanats would not be influenced by establishing a new well.

Comment:

Please also note the supplement to this comment: https://www.hydrol-earth-syst-sci-discuss.net/hess-2019-407/hess-2019-407-RC3- supplement.pdf.

Response:

We appreciate these comments. The comments were considered and modified.

Please also note the supplement to this comment:
https://www.hydrol-earth-syst-sci-discuss.net/hess-2019-407/hess-2019-407-AC4-supplement.pdf